# The Role of a New Compound Micronutrient Multifunctional Fertilizer against *Verticillium dahliae* on Cotton

**DOI:** 10.3390/pathogens10010081

**Published:** 2021-01-19

**Authors:** Yalin Zhang, Lihong Zhao, Zili Feng, Hongfu Guo, Hongjie Feng, Yuan Yuan, Feng Wei, Heqin Zhu

**Affiliations:** 1State Key Laboratory of Cotton Biology, Institute of Cotton Research, Chinese Academy of Agricultural Sciences, Anyang 455000, China; zhangyalin@caas.cn (Y.Z.); zhaolihong@caas.cn (L.Z.); fengzili@caas.cn (Z.F.); fenghongjie@caas.cn (H.F.); yuanyuan06@caas.cn (Y.Y.); 2Puyang City Academy of Agricultural Sciences, Puyang 457000, China; guohongfu@caas.cn; 3State Key Laboratory of Cotton Biology, Zhengzhou Research Base, Zhengzhou University, Zhengzhou 450000, China

**Keywords:** cotton, *Verticillium dahliae*, pathogenicity, micronutrient, prevention strategy

## Abstract

*Verticillium dahliae* Kleb., the causal pathogen of vascular wilt, can seriously reduce the yield and quality of many crops, including cotton (*Gossypium hirsutum*). To control the harm caused by *V. dahliae*, considering the environmental pollution of chemical fungicides and their residues, the strategy of plant nutrition regulation is becoming increasingly important as an eco-friendly method for disease control. A new compound micronutrient fertilizer (CMF) found in our previous study could reduce the damage of cotton Verticillium wilt and increase yield. However, there is little information about the mode of action of CMF to control this disease. In the present study, we evaluated the role of CMF against *V. dahliae* and its mechanism of action in vitro and in vivo. In the laboratory tests, we observed that CMF could inhibit hyphal growth, microsclerotia germination, and reduce sporulation of *V. dahliae*. Further studies revealed that the biomass of *V. dahliae* in the root and hypocotyl of cotton seedlings treated with CMF were significantly reduced compared with the control, and these results could explain the decline in the disease index of cotton Verticillium wilt. Furthermore, those key genes involved in the phenylpropanoid metabolism pathway, resistance-related genes defense, and nitric oxide signaling pathway were induced in cotton root and hypocotyl tissue when treated with CMF. These results suggest that CMF is a multifaceted micronutrient fertilizer with roles in inhibiting the growth, development, and pathogenicity of *V. dahliae* and promoting cotton growth.

## 1. Introduction

*Verticillium dahliae* Kleb. is a devastating soilborne fungal pathogen with worldwide distribution, which threatens agricultural production in the long term. It is able to infect more than 200 known plant species besides cotton in the world [1,2,3]. Cotton, a primary natural fiber crop, is of great importance to the global textile industry [4]. Unfortunately, China suffers from tremendous yield losses of cotton in *V. dahliae*-infested soil annually [5]. Much research has been made to understand the interaction between cotton-*V. dahliae* and to control Verticillium wilt [6], such as breeding resistant cultivars [7], crop rotation [8], developing soil fumigants [9], and the use of chemical fungicides [10], as well as biological control [3]. Unfortunately, because of the scarcity of resistance in host germplasm, there are no highly resistant cotton cultivars [11]. Crop rotation cannot be widely promoted due to limited land resources [12]. The control effect of a biological control agent is unstable because of its dependence on the soil environment and climate [13]. To date, chemical fungicides have to be selected as the principal method to control *V. dahliae* in agricultural production [1]. However, long-term use of chemical fungicides not only poses a threat to the ecological environment and human health but also improves the resistance of *V. dahliae* [14]. Therefore, there is an urgent need to develop environmentally-friendly strategies for effective control of Verticillium wilt disease.

Up to now, micronutrients have been confirmed to have the potential to inhibit soilborne root pathogens on plants, and optimization of plant micronutrients is a sustainable approach to improve crop health [10,15,16]. Micronutrients not only affect the normal growth and development of plants but also directly or indirectly influence the susceptibility and disease resistance of plants in many ways [17]. Specific single micronutrients, such as Fe, Zn, Mn, Cu, Se, Si, have been proven to have a suppressive effect on many plant diseases [18,19,20,21]. The application of Zn micronutrient could significantly reduce the occurrence of winter wheat (*Triticum aestivum*) leaf rust (*Puccinia recondita*), and lower the symptoms of hop (*Humulus lupulus*) leaf roll disease, and effectively prevent corn (*Zea mays*) stalk rot (*Fusarium*) [22]. As an inorganic fungicide, Cu is a component of phenolase, which directly participates in the metabolism of phenols in plants, affects the synthesis of phytoalexins, and inhibits the germination and growth of fungal spores. Evidence demonstrated that spraying Cu micronutrient lowered leaf blight of gourd (*Lagenaria siceraria*) [23]. Likewise, Mn participates in the production of phenolic compounds and reduces the incidence of various diseases, such as poplar (*Populus tomentosa*) canker (*Lonsdalea populi*), soybean (*Glycine max*) virus disease, cabbage (*Brassica oleracea*) root swelling (*Plasmodiophora brassicae*), and cotton Verticillium wilt [24]. A reasonable application of multiple micronutrients can induce plant systemic resistance. It is of great significance to reduce pesticide consumption and environmental pollution [25]. A study found that the application of P, K, Zn fertilizers combined with the low-level application of N and Fe could significantly reduce pea (*Pisum sativum*) powdery mildew disease [26]. Besides, the combined application of Zn and Se fertilizers could increase the GSH-Px activity of millet (*Panicum miliaceum*) more than a single application and had a significant synergistic effect on the antioxidant enzyme activity of millet leaves and the yield [27]. Regular application of Mo and Ca fertilizers to tobacco (*Nicotiniana tabacum*) enhanced the defense of tobacco against bacterial wilt (*Ralstonia solanacearum*) and improved its resistance to bacterial wilt [21]. In summary, as a supplement to chemical fungicides and biological control measures, the combination of specific micronutrients can aid the control of plant diseases, increase crop yields, and obtain significant economic benefits.

Compound micronutrient fertilizer (CMF) was jointly developed by the Institute of Cotton Research of the Chinese Academy of Agricultural Sciences and the Puyang City Academy of Agricultural Sciences, produced by Puyang City Nongke Agrochemical Co., Ltd. whose merchandise name is Mianweike and the registration number is agricultural fertilizer (2009) No. 1406, and the main components are Cu, Zn, Mn micronutrients with a content of more than 10%. In our previous study, CMF was used to prevent and control cotton Verticillium wilt by spraying or drip irrigation, with a control effect of over 40% and an increase in yield of 7.0~9.3% in Alaer and Shihezi, Xinjiang, China [28]. However, there is little information about the mechanism of action of CMF to control cotton Verticillium wilt.

The objectives of this work were to analyze the effect of CMF on mycelial growth, microsclerotia germination, and sporulation of *V. dahliae* to evaluate the direct impact of CMF on the pathogen. Furthermore, we studied the expression of resistance-related genes in cotton treated with CMF to illuminate the induced resistance of cotton. Revealing these effects are of great importance to understand better the mechanism of the action of CMF against *V. dahliae* on cotton.

## 2. Results

### 2.1. Inhibitory Effect of CMF on Mycelial Growth of V. dahliae

To analyze the function of CMF in fungal growth and microsclerotia production, *V. dahliae* strain Vd080 was grown on potato dextrose agar (PDA). At 9 days post inoculation (dpi), all treatments of different concentrations of CMF led to a significant inhibition in hyphal growth and a significant reduction in melanin and microsclerotia production of *V. dahliae* strain Vd080 (Figure 1a). The growth inhibition rates of CMF on *V. dahliae* strain Vd080 were related to the concentration of CMF, with the respective inhibition rates of 25.57%, 37.50%, 49.63%, 59.63%, and 68.01%, corresponding to the concentrations of 0.16, 0.31, 0.63, 2.50, and 10.00 g/L (Table 1). In addition, the regression equation was Y = 0.5738 X + 4.9727, R^2^ = 0.9477 (*p <* 0.01). EC_50_ and EC_95_ values of CMF were 1.12 g/L and 820.58 g/L, respectively.

### 2.2. Inhibition of CMF on Microsclerotia Germination of V. dahliae

With the extension of culture time, the germination rates of microsclerotia gradually increased. Compared with the untreated control, the germination rates of microsclerotia treated with CMF were significantly lower. At 72 h post treatment (hpt), the microsclerotia germination rates of *V. dahliae* strain Vd080 treated with CMF concentrations of 1 g/L and 5 g/L were 38.33% and 18.93%, respectively, with the inhibition ratios being 40.85% and 70.79%, respectively (Figure 1b).

### 2.3. Influence of CMF on Sporulation of V. dahliae

As expected, CMF also had a pronounced inhibitory effect on the conidia production of *V. dahliae*. Regardless of 1 g/L or 5 g/L concentration of CMF treatment, sporulation of *V. dahliae* was significantly lower than the control. At 72 hpt, the spore yields of *V. dahliae* treated with CMF concentrations of 1 g/L and 5 g/L were 12.20 × 10^6^ spores/mL and 9.20 × 10^6^ spores/mL, with the reduction rates of 38.38% and 53.53% compared to the control (19.80 × 10^6^ spores/mL), respectively (Figure 1c).

### 2.4. Suppression of Cotton Verticillium Wilt in the Greenhouse

In the greenhouse trials, cotton plants treated with 1 g/L of CMF reduced wilt development when assessed 24 days post spraying treatment (Figure 2a). The xylem of CMF-treated plants showed lighter vascular browning than that of the control (Figure 2b). The disease index (DI) of 17.44 for the CMF-treated plants, compared to 31.89 for the control. The control efficacy was 62.33%, 46.20%, and 45.31% when assessed at 10, 17, and 24 days post spraying CMF (Figure 2c).

In addition, the qPCR results showed that the biomass of *V. dahliae* strain Vd080 in roots of infected cotton plants increased first and then decreased, with the peak at 48 h post treatment (hpt). At 48 hpt, the biomass of *V. dahliae* strain Vd080 in CMF-treated cotton plants only accounted for one-tenth of the untreated control (Figure 3a). In hypocotyl tissue, the *V*. *dahliae* biomasses with or without CMF treatment all increased over time, but the increase in CMF-treated cotton plants was slower than that of the control plants. At 96 hpt, the *V. dahliae* strain Vd080 biomass of the control was 4.26-fold compared to CMF treatment (Figure 3b). Collectively, these results proved that CMF has an inhibitory effect on the infection, colonization, and spread of *V. dahliae* on cotton.

### 2.5. Effect of CMF Treatment on Cotton Seedling Development

There was no significant difference between the two treatments in root length and plant height when assessed 24 days after treatment, compared to the control (Table 2). Interestingly, the fresh weight of cotton seedlings was significantly increased by 14.53% compared to the control (Table 2).

### 2.6. Expression Analysis of Resistance-Related Genes

Gene expression patterns varied over time and had specific expression in different plant tissues. As key genes of the lignin metabolism pathway, the genes of peroxidase, *POD;* polyphenol oxidase, *PPO*; and phenylalanine ammonia lyase, *PAL*, were strongly upregulated in both root and hypocotyl tissues of cotton treated with CMF, compared to the control. In detail, the expression of *POD* in root and hypocotyl tissues reached its peak at 12 hpt and 24 hpt, with the respective 1.7-fold and 2.1-fold compared to the control. Subsequently, its expression level declined (Figure 4a). The expression kinetics of *PPO* and *PAL* genes were similar to *POD*. It was worth mentioning that the expression level of *PPO* in hypocotyl tissue increased 55 times on average compared with those in root tissue. In addition, the gene expression level of *PPO* in hypocotyl tissue of cotton treated with CMF was significantly higher than the control, with a 1.7-fold compared to the control at 12 hpt (Figure 4b). The expressions of *POD*, *PPO*, and *PAL* were all significantly upregulated in both root and hypocotyl tissues of those plants treated with CMF within 48 hpt compared with control plants (Figure 4c).

As a broad-spectrum resistance function gene, *PR10*; *PR10* expression was upregulated in hypocotyl tissue with a trend of fall and rise, then fall and rise. This phenomenon occurred in both cotton treated or untreated with CMF. However, the expression of *PR10* in root tissue of cotton treated with CMF was rise and fall, then rise and fall, opposite to the control, and with the highest expression of 11.6-fold to the control at 48 hpt in root (Figure 4d). Transcription levels of basic chitinase, *CHI,* and cadinene synthase, *CAD* genes were also upregulated at 24 hpt in roots treated with CMF, with respective increases of 2.8-fold and 1.1-fold compared to the control (Figure 4e,f). Similarly, the expression levels significantly increased by 5.0-fold and 7.9-fold over them in roots, respectively.

As key genes of the phenylpropanoid metabolic pathway, 4-coumaric acid-CoA ligase, *4CL*, and cinnamic acid hydroxylase, *C4H1* expression was drastically upregulated in CMF-treated cotton root during 12 hpt. Subsequently, their expression levels decreased significantly below those of mock-treated roots at 24 hpt, and their peak expression levels were 1.6 times and 8.6 times over the control, respectively. In CMF-treated cotton hypocotyl tissue, the expression levels of *4CL* and *C4H1* were not significantly different from those of the control. Until 48 hpt, their expression increased and was significantly higher than the control (Figure 4g,h).

*NOA1* (Nitric oxide associated 1) involved in protein translation in the chloroplast has been indirectly linked to nitric oxide (NO) accumulation. Regardless of whether the cotton was treated or untreated with CMF, the expression level of *NOA1* in root and hypocotyl tissues was downregulated. Furthermore, in cotton root treated with CMF, the expression of *NOA1* was significantly downregulated, with about one-half of the control at 6 hpt (Figure 4i).

### 2.7. Suppression of Cotton Verticillium Wilt in the Field

For the CMF treatment, the DI values ranged from 6.50 to 16.61, compared to corresponding values for the control from 17.60 to 35.52, with a significant difference level, accompanied by the control efficacies of CMF from 63.05% to 53.24% (Figure 5). There were no plant samples with wilt scores of 3 and 4 in CMF treatment. This result demonstrates that CMF could effectively curb the further deterioration of diseased cotton plants.

Cotton yield results showed that the seed cotton weight of CMF treatment reached 2770.17 kg/ha, increasing 7% compared to the control (2588.82 kg/ha). The increase in lint cotton yield was roughly the same as that of seed cotton, with an increase of 8.19%. As an important element of cotton yield, single cotton boll weight treated with CMF increased by 6.75% compared to the control, but with no significant difference. Likewise, there was no significant difference in cotton lint percentage between the two treatments (Table 3).

## 3. Discussion

Micronutrients are necessary for the normal growth and development of plants. Reasonably balanced fertilization can make plants grow vigorously and enhance disease resistance [29]. In recent years, more and more attention has been paid to the harm of chemical fungicides on the environment and ecological balance [30]. Mounting evidence shows that some special micronutrients have the potential to control plant diseases [15,18]. Micronutrient elements, such as Cu and Zn, are widely used in inorganic fungicides to directly resist the infection of pathogenic fungi, reduce the germination of spores, and the growth of pathogenic fungi [16,31]. In this study, we determined that CMF (the main components include Cu, Zn, Mn micronutrients) could inhibit mycelial growth and sporulation of *V. dahliae* (Figure 1). Furthermore, the regression equation of CMF was calculated, with EC_50_ and EC_95_ values of 1.12 g/L and 820.58 g/L, respectively. These results will provide guidance for the prevention and control of cotton Verticillium wilt in production. A current study suggests that Se can reduce the pathogenicity of *Sclerotinia sclerotiorum* by inhibiting sclerotial formation and germination [10]. Besides, growing research show that micronutrients could promote plant growth, heighten biomass, and increase crop yield [15,16,32]. Similarly, this study suggests that the fresh weight of cotton seedlings treated with CMF was significantly increased by 14.53%, and lint cotton yield increased by 8.19%. Collectively, these results indicate that CMF could not only directly inhibit the growth of *V. dahliae* but also increase cotton biomass.

As the long-term survival resting structures of *V. dahliae*, microsclerotia play a critical role in the disease cycle as primary sources of infection [33]. The synthesis of microsclerotia and accumulation of melanin are directly coupled with the pathogenicity or virulence of *V. dahliae*, which are always considered important targets for Verticillium wilt control [34]. Unfortunately, the microsclerotia have a strong ability to resist adversity and can survive for more than 10 years in the field soil [1,35]. Hence, inhibiting microsclerotia formation, on the one hand, can directly reduce the initial infection; on the other hand, lower the germination of hyphae, which is considered an effective approach to control *V. dahliae.* In this study, the formations of microsclerotia and melanin in *V. dahliae* were almost completely inhibited after CMF treatment. Furthermore, CMF also has an inhibitory effect on the microsclerotia germination of *V. dahliae* (Figure 1b). This agrees with previous findings that Se inhibits mycelial growth and sclerotial formation of *S. sclerotiorum* [10]. These findings show that CMF may reduce the pathogenicity of *V. dahliae* by inhibiting the formation of microsclerotia.

Verticillium wilt is caused by the systemic colonization of *V. dahliae* in plant vascular tissues, and plant roots usually serve as the first battleground between *V. dahliae* and plant host [36]. The colonization of *V. dahliae* on cotton roots directly affects the incidence of cotton Verticillium wilt [37]. Next, the xylem is the second battlefield of *V. dahliae* and plant host that compete for nutrition [38]. As a new type of pathogen antagonist, phosphate can reduce the colonization and spread of a variety of pathogenic fungus in plants [39]. Similarly, CMF could reduce the colonization of *V. dahliae* in cotton root and the spread in hypocotyl by detecting the biomass of *V. dahliae* in cotton tissues (Figure 3), resulting in severely impaired virulence on cotton (Figure 2a). Even in the field that was previously infested with Verticillium wilt, this new micronutrient fertilizer significantly reduced wilt development, and the suppressive effect was more than 53% and could last up to 42 days (Figure 5). These findings suggest that CMF not only improves the ability of cotton to resist the infection of *V. dahliae*, but also reduce pathogenicity of *V. dahliae*.

In general, under biological stress conditions, reactive oxygen species (ROS) are widely found in organisms, which play an important role in fighting against pathogens [16,40]. After plant recognition of pathogen attack, a rapid oxidative burst of oxygen radicals is needed to trigger plant defense mechanisms and induce *POD*, *PPO*, *SOD* antioxidant enzyme genes to participate in the regulation of oxygen radicals [41]. Besides, *POD*, *PPO* together with *PAL*, *4CL,* and *C4H1* participate in the phenylpropanoid metabolic pathway, playing an important role in the plant defense response [42,43]. The enhanced expression of defense-related genes contributes to the activation of plant defense systems [44]. Similarly, foliar spraying with phosphate can not only inhibit cucumber (*Cucumis sativus*) powdery mildew (*Podosphaera xanthii*) by increasing the activity of related defense enzymes but also improve cucumber’s local and systemic resistance to powdery mildew and other diseases [45]. In our study, the expression levels of *POD*, *PPO*, *PAL*, *4CL,* and *C4H1* genes in the root and hypocotyl tissues of cotton seedlings treated with CMF were all upregulated, but the levels of the gene expression vary in different tissues, especially the expression level of *PPO* in the hypocotyl was 55 times higher than in the root (Figure 4). Therefore, these results indicate that CMF could induce cotton to acquire local and systemic resistance by increasing the expression of key genes of the phenylpropanoid metabolic pathway to inhibit the harm of *V. dahliae*. In addition, many disease-resistant genes were induced by micronutrients against pathogen infection [46,47]. In this study, we provide evidences that CMF induces a varying degree increase of resistance-related genes transcripts, including *PR10*, *CHI,* and *CAD*. It is noteworthy that the nitric oxide (NO) signaling pathway co-functions with the ROS signaling pathway in plant biotic interactions [48]. The expression of *NbNOA1* was downregulated because of a significant decrease in NO accumulation [49]. In the present study, *NOA1* was also downregulated due to CMF treatment reducing NO accumulation. In summary, we may conclude that the mechanism of action of CMF induced cotton to respond to the invasion of *V. dahliae* is complicated, involving the phenylpropanoid metabolism pathway, resistance-related genes defense, and NO signaling pathway. Furthermore, CMF is also promoting plant growth and development. Further research is needed to investigate whether the defense induced by CMF depends on the natural defense mechanism of plants.

## 4. Materials and Methods

### 4.1. Cotton Cultivar, Fungal Strain and Culture Conditions

The cotton cultivar used in these tests was Lumianyan 21, which is tolerant to *V. dahliae*. The *V. dahliae* strain Vd080, which was isolated from cotton collected in Xinji, Hebei, China, was single-spore isolated and stored at −80 °C in 20% (*v*/*v*) glycerol. Cultures were activated on a potato dextrose agar medium to observe biological characteristics. To induce conidia formation for the infection assays, isolates were incubated in liquid Czapek medium at 25 °C with shaking at 150 r/min [50].

### 4.2. Preparation of V. dahliae Fermentation Broth

The *V. dahliae* strain Vd080 was inoculated in liquid Czapek medium, cultured on a constant temperature shaker, 150 r/min, 25 °C for 5–7 days. Next, *V. dahliae* strain Vd080 sporulation was calculated using a hemocytometer (QIUJING, Shanghai, China) and diluted with distilled water to 1 × 10^7^ CFU/mL [51].

### 4.3. Assessing the Effect of CMF on the Growth Rate of V. dahliae

A mycelial disk of *V. dahliae* strain Vd080 (4 mm diameter) was placed in the center of the PDA medium plate, which contained gradient configuration (0, 0.16, 0.31, 0.63, 2.50, and 10.00 g/L) of CMF. The plate was then immediately sealed with plastic film and incubated in the dark at 25 °C for 9 days. Five repeats were set for each concentration of CMF. The *V. dahliae* colony diameter was measured on 7- and 9-days post inoculation. The growth inhibition rate (%) = (control colony diameter − treated colony diameter)/control colony diameter × 100, and the EC_50_ and EC_95_ values of CMF were calculated [42].

### 4.4. Effect of CMF on the Microsclerotia Germination of V. dahliae

The culture of microsclerotia and preparation its suspension were performed as described previously [52]. Microsclerotia suspension (100 μL) and the same amount of liquid Czapek medium containing different concentrations of CMF were mixed and cultured at 20 °C in the dark. The germination rates of microsclerotia at 24, 48, and 72 h post treatment were estimated by a microscope (Toshiba, Tokyo, Japan) with a 40× eyepiece. The numbers of 100 microsclerotia were observed each time, and the length of the germ tube of the microsclerotia exceeded half regarded as germination.

### 4.5. Assessing the Effect of CMF on the Sporulation of V. dahliae

Spore concentration of *V. dahliae* strain Vd080 was adjusted to 1 × 10^7^ CFU/mL, then 5 μL of the spore suspension was added to liquid Czapek medium with CMF concentrations equal to 1 g/L and 5 g/L, and incubated on a constant temperature shaker at 25 °C with shaking at 150 r/min. The experiment was performed five times. Spore yields at 24, 48, and 72 hpt were estimated with a hemocytometer [37].

### 4.6. Suppressive Effect of CMF on V. dahliae in the Greenhouse

Cotton seeds were sown in paper pots (6 cm in diameter and 10 cm in height, without a bottom) filled with an autoclaved substrate (vol/vol, vermiculite/sand = 6/4). The inoculum concentration of *V. dahliae* strain Vd080 was adjusted to 1 × 10^7^ CFU/mL and inoculated onto 25–30 cotton seedlings at the first euphylla stage by immersing the seedlings’ roots in the *V. dahliae* strain Vd080 conidial suspension [53]. The experiment was performed three times. The cotton seedlings began to sporadically exhibited chlorosis and wilting at 4 days post inoculation. Meanwhile, CMF was diluted with sterile deionized water (SDW) to the concentration 1 g/L. The cotton seedlings were sprayed with CMF suspension via a handheld sprayer. Control plants were sprayed with SDW. The cotton seedlings were cultivated in a standard greenhouse at 25–30 °C under a 16-h/8-h light-dark photoperiod. Disease progress was recorded 10, 17, and 24 days post spraying treatment. The classification of Verticillium wilt was as follows: 0 = healthy plant; 1 = one or two cotyledons showing symptoms; 2 = both cotyledons and one true leaf showing symptoms; 3 = both cotyledons and two true leaves showing symptoms; 4 = plant died. The disease index was calculated as follows: DI = (0n_0_ + 1n_1_ + 2n_2_ + 3n_3_ + 4n_4_)/4n) × 100, where n_0_–n_4_ were the numbers of plants with each of the corresponding disease ratings, and n was the total number of plants assessed [53]. Meanwhile, the cotton seedlings’ biological indicators, including root length, plant height, and fresh weight, were measured at 24 dpt.

### 4.7. qPCR Quantification of Fungal Biomass in Plant Tissue

To quantify pathogen colonization level in root and hypocotyl tissues treated with CMF, *V. dahliae* strain Vd080 biomass on cotton plants, it was estimated by quantitative PCR. The root and hypocotyl tissues of cotton were harvested by cutting with a sharp scissor at 6, 12, 24, 48, and 96 h post spraying treatment (hpt) as described above. Plant tissues were sterilized for 10 min with 70% ethanol and rinsed three times with sterile water, five cotton plants as a treatment. Fungi-plants mixed DNA was extracted using the hexadecyltrimethylammonium bromide method (CTAB), and DNA concentration of each sample was measured by NanoDrop 2000 (Thermo Scientific Corporation, Beijing, China) and adjusted to 50 ng/μL for the qPCR reaction [54]. Primers Vdβt-F and Vdβt-R were designed based on the *V. dahliae* β-tubulin gene [55]. The endogenous control *G. hirsutum* actin gene was amplified using primer act-F and act-R [37] (Table 4). Cycle thresholds were determined in three biological replicates per sample using LightCycler 480 system (Roche Diagnostics, Mannheim, Germany) and the TOP SYBR Green qPCR SuperMix (TransGEN, Beijing, China) as the reporter dye. Real-time qPCR was run for 45 cycles. Initial denaturation and enzyme activation were carried out at 94 °C for 30 s, denaturation at 95 °C for 5 s, annealing at 60 °C for 15 s, and elongation at 72 °C for 10 s.

### 4.8. Expression Analysis of Resistance-Related Genes by qRT-PCR

The relative transcript levels of resistance-related genes were determined with a quantitative reverse transcription PCR (qRT-PCR) method. The root and hypocotyl tissues of cotton were harvested at 6, 12, 24, 48, and 96 hpt, as mentioned above. Total RNA was extracted from cotton Lumianyan 21 using an RNAprep Pure Plant Kit (Tiangen, Beijing, China) and reversely transcribed into cDNA using the PrimeScript™ II 1st Strand cDNA Synthesis Kit (Takara, Japan). The *G. hirsutum* Ubiquitin gene was used as an endogenous reference for the normalization of cotton mRNA [56]. qRT-PCR was performed to quantify the transcript levels of the following genes: three key genes in the lignin metabolism pathway (peroxidase, *POD;* polyphenol oxidase, *PPO*; and phenylalanine ammonia lyase, *PAL*), three pathogenesis-related (PR) genes (disease-related protein gene, *PR10*; basic chitinase, *CHI*; and cadinene synthase, *CAD*); two core genes in the phenylpropanoid metabolic pathway (4-coumaric acid-CoA ligase, *4CL*, and cinnamic acid hydroxylase, *C4H1*) and an important gene responding to *NO* signaling pathway (nitric oxide associated 1, *NOA1*) with specific primers (Table 4).

### 4.9. Control Efficacy of CMF Against Cotton Verticillium Wilt and Increasing Yield in the Field

Field experiments were carried out in the artificial disease nursery located in Anyang, Henan, China (36°05′19.46″ N, 114°30′47.21″ E). The field has more than 20 years of cotton crops and is to a serious Verticillium wilt field. Cotton (cv. Lumianyan 21) seeds were sown manually on 30 April 2020. The planting pattern was 80 cm equal row with 26 cm plant spacing and a density of 48,000 plants/ha. There were two treatments, including spraying CMF or not. Each replicate plot had 6 rows of 8 m in length, containing about 200 cotton plants. Both treatments were repeated three times. CMF was sprayed twice on 29 June and 15 July by using an artificial sprayer at a concentration of 1 g/L, the same amount of water as the control. Symptoms of Verticillium wilt on cotton were investigated at 11, 25, and 42 days post the first spraying CMF. Every cotton plant was listed, and the Verticillium wilt disease grade was continuously recorded. The classification of Verticillium wilt was as follows: 0 = no symptoms, 1 = ≤33%, 2 = >33% and ≤66%, 3 = >66% and ≤99%, and 4 = >99% leaves with chlorosis wilt symptoms [53]. On 30 September, each plot was manually harvested, and the cotton yield was measured, including seed cotton weight, lint cotton weight, and single boll weight.

### 4.10. Statistical Analysis

The mean and standard error (SE) of all results were calculated, and one-way ANOVA and Tukey’s honestly significant difference (Tukey’s HSD) tests were performed using GraphPad Prism 8 to generate *p*-values. Significant differences are indicated with asterisks: * *p* < 0.05 and ** *p* < 0.01.

## Figures and Tables

**Figure 1 pathogens-10-00081-f001:**
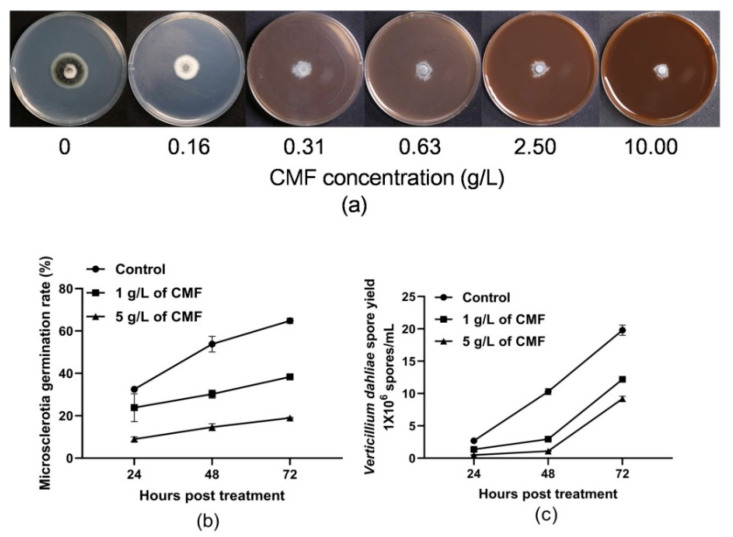
Influence of compound micronutrient fertilizer (CMF) on phenotypic characterization of *V. dahliae*. (**a**) Colony morphology of *V. dahliae* treated with CMF (0, 0.16, 0.31, 0.63, 2.50, 10.00 g/L) on potato dextrose agar (PDA) at 9 days post inoculation (dpi). (**b**) Inhibition of CMF on microsclerotia germination of *V. dahliae*. (**c**) Effect of CMF on sporulation of *V. dahliae*. Means and standard errors were calculated from five independent experiments according to Tukey’s honestly significant difference (HSD) test.

**Figure 2 pathogens-10-00081-f002:**
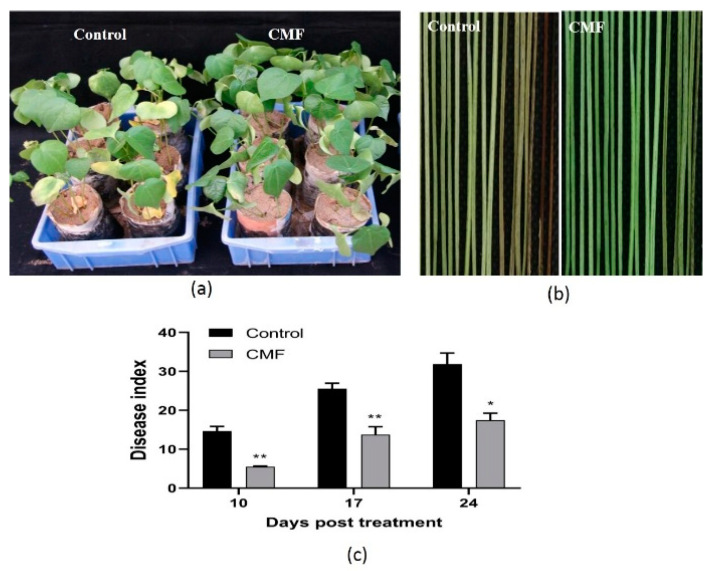
CMF protects cotton plants from *V. dahliae* infection in the greenhouse. (**a**) Cotton plants at 24 days post treatment with CMF after inoculation with *V. dahliae*. (**b**) Vascular browning found in xylem from infected cotton plants at 24 dpt. (**c**) Disease symptom progress from three replicate experiments in the greenhouse assay. Asterisks indicate statistically significant differences compared with control (* *p* < 0.05 and ** *p* < 0.01).

**Figure 3 pathogens-10-00081-f003:**
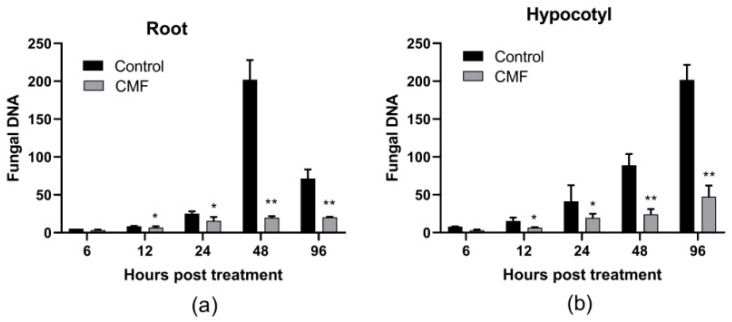
Detection of fungal biomass in infected cotton tissues. (**a**) Quantification of fungal DNA in root section. (**b**) Quantification of fungal DNA in hypocotyl tissue. Asterisks indicate statistically significant differences compared with control (* *p* < 0.05 and ** *p* < 0.01).

**Figure 4 pathogens-10-00081-f004:**
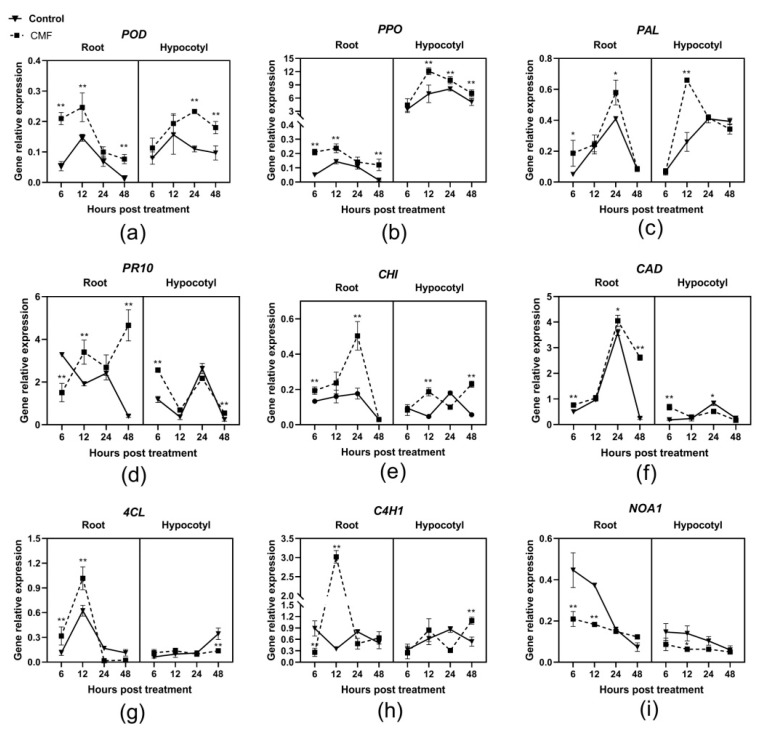
Induction of resistance-related gene expression in cotton treated with CMF after inoculation with *V. dahliae*. Samples of cotton root and hypocotyl tissues were collected 6, 12, 24, and 48 h post treatment. Data are means of three replicate experiments in the greenhouse assay. The bars represent the average induction (± SE) of gene transcripts normalized to the Ubiquitin gene for three replicates. Asterisks indicate statistically significant differences compared with control (* *p* < 0.05 and ** *p* < 0.01). (**a**) polyphenol oxidase, *POD*, (**b**) polyphenol oxidase, *PPO*, (**c**) phenylalanine ammonia lyase, *PAL*, (**d**) *PR10*, (**e**) basic chitinase, *CHI*, (**f**) cadinene synthase, *CAD*, (**g**) 4-coumaric acid-CoA ligase, *4CL*, (**h**) cinnamic acid hydroxylase, *C4H1*, (**i**) nitric oxide associated 1, *NOA1*.

**Figure 5 pathogens-10-00081-f005:**
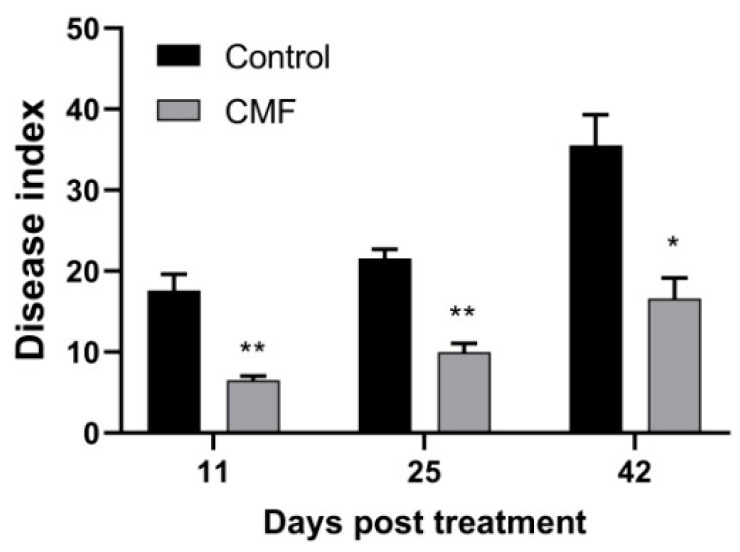
CMF protects cotton plants from *V. dahliae* infection in the field. Data are means of three replicate experiments in the field assay. Asterisks indicate statistically significant differences compared with control (* *p* < 0.05 and ** *p* < 0.01.).

**Table 1 pathogens-10-00081-t001:** Inhibitory effect of compound micronutrient fertilizer (CMF) on mycelial growth of *V. dahliae*.

CMF Concentration (g/L)	Colony Diameter (mm)	Growth Inhibition Rate (%)
0	27.20 ± 1.69 a	/
0.16	19.70 ± 1.77 b	27.57
0.31	17.00 ± 1.05 c	37.50
0.63	13.70 ± 0.67 d	49.63
2.50	10.90 ± 0.74 e	59.93
10.00	8.70 ± 0.82 f	68.01

The data represent the mean ± SE (n = 5). Data followed by different letters indicate statistically significant differences at the level of *p <* 0.05 according to Tukey’s HSD.

**Table 2 pathogens-10-00081-t002:** Effect of CMF on cotton biomass.

Treatment	Root Length (cm)	Plant Height (cm)	Fresh Weight (g)
Control	9.43 ± 0.44 a	11.74 ± 0.61 a	1.07 ± 0.03 b
CMF	9.53 ± 0.11 a	11.35 ± 1.18 a	1.23 ± 0.06 a

Means ± SE (n = 3) labeled with different letters indicate a significant difference (*p* < 0.05) according to Tukey’s HSD test.

**Table 3 pathogens-10-00081-t003:** Effect of CMF on cotton yield parameters.

Treatment	Seed Cotton (kg/ha)	Lint Cotton (kg/ha)	Lint Percentage (%)	Single Boll Weight (g)
Control	2588.82 ± 64.61 b	1043.24 ± 43.92 b	40.34 ± 2.69 a	4.79 ± 0.23 a
CMF	2770.17 ± 46.69 a	1128.70 ± 24.82 a	40.74 ± 0.21 a	5.13 ± 0.15 a

Means ± SE (n = 3) labeled with different letters indicate a significant difference (*p <* 0.05) according to Tukey’s HSD test.

**Table 4 pathogens-10-00081-t004:** qPCR detection of relative gene expressions.

Gene Name	Primer Sequence (5′–3′)
qPCR quantification of fungal biomass	
*Vdβt*	F: AACAACAGTCCGATGGATAATTC
	R: GTACCGGGCTCGAGATCG
*actin*	F: CCTATGTTGCCCTGGACTATGAGC
	R: GGACAACGGAATCTCTCAGCTCC
qRT-PCR detection of genes relative expression	
*POD*	F: CCGCATAACCATCACAAG
	R: ACTCTCATCACCTTCAACA
*PPO*	F: ATATCCTTGTTCTGTCTGCTA
	R: CTCCTTCTACCGTCTCTTC
*PAL*	F: TGGTGGCTGAGTTTAGGAAA
	R: TGAGTGAGGCAATGTGTGA
*PR10*	F: ATGATTGAAGGTCGGCCTTTAGGG
	R: CAGCTGCCACAAACTGGTTCTCAT
*CHI*	F: CTTAGCCCAAACTTCCCA
	R: TACATTGAGTCCACCGAGAC
*CAD*	F: TAACAACAATGATGCCGAGAA
	R: ATGGTCCAAAGATGCTACTGC
*4CL*	F: ATTCAAAAGGGAGATGCC
	R: GAGAAGGGCAAAGCAACA
*C4H1*	F: CCGAACCCGACACCCATAAGC
	R: GCAGGGATGTCATACCCACCAAG
*NOA1*	F: GAGGATGCTGAAAGACCTGCTA
	R: TCTCAACTGGCTTGGGTACATG
*Ubiquitin*	F: GAGTCTTCGGACACCATTG
	R: CTTGACCTTCTTCTTCTTGTGC

## Data Availability

The data used to support the findings of this study are available from the corresponding author upon request.

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
