# Peer review of "The Role of a New Compound Micronutrient Multifunctional Fertilizer against Verticillium dahliae on Cotton"

_pathogens, 2021, doi:10.3390/pathogens10010081_

Round 1

Reviewer 1 Report

The authors provide an results from an ongoing study on the effect of compound micronutrient fertilizer (CMF).  The results seemed promising and may provide useful details on the mechanism of action.  I was concerned that the concentrations used in the study (in vitro) were very high and the inhibitory effects may not translate to field situations (Fig 1).  However, the effects on fungal physiology were interesting and worthy of publication.  The research was conducted using a proprietary formulation and only sketchy details of its composition are provided.  It is therefore difficult to assess the meaning of the results and this greatly limits the usefulness of the work.  For example, the data cannot be cited to show micronutrient effects because the composition is unknown and therefore the work work has a lower impact.

A file with handwritten notes is attached

Author Response

Comment 1: I was concerned that the concentrations used in the study (in vitro) were very high and the inhibitory effects may not translate to field situations (Fig 1). However, the effects on fungal physiology were interesting and worthy of publication.

Our response: First of all, thank you for your meticulous suggestion. We have revised and highlighted them in the whole manuscript. In fact, in laboratory and field trials, we have conducted a large number of tests on the concentration of CMF in order to determine the safe and effective concentration to control cotton verticillium wilt. Finally, 1 g/L concentration of CMF is recommended to use in field production, with the control efficacy from 53.24% to 63.05%.

Comment 2: The research was conducted using a proprietary formulation and only sketchy details of its composition are provided. It is therefore difficult to assess the meaning of the results and this greatly limits the usefulness of the work. For example, the data cannot be cited to show micronutrient effects because the composition is unknown and therefore the work work has a lower impact.

Our response: Taking into account trade secret and the interests of CMF sales company, we apologize for failing to disclose more detailed content of Cu, Zn and Mn micronutrients in CMF, the mass ratio of Cu, Zn and Mn micronutrients is 3:3:4. which is the core protection of this proprietary formulation. To be honest, CMF formulation is obtained after a lot of experiments and has been widely used in cotton field in China. Because of its good control effect on cotton Verticillium wilt, it has been purchased by a large number of farmers every year. Hope to get your understanding and thanks again.

Reviewer 2 Report

All my comments and suggestions for authors could be find in the attached file.

Author Response

Comment 1: Mention Latin name of cotton also in this section.

Our response: Yes, we have added it (line 18).

Comment 2: I suggest to mention all the Latin names when any of living organisms are mentioned for the first time. I mean here cotton as well as all other plants, which are widely mentioned, and causal agents of all different diseases mentioned in the article, e.g., poplar canker, cabbage root swelling etc. This note refers not only to Introduction, but also to other sections of the article.

Our response: Many thanks for your careful reading. In the whole manuscript, we have added the Latin names of all plants and pathogens when they first mentioned.

Comment 3: The demand of the Journal is “Finally [in Introduction], briefly mention the main aim of the work and highlight the main conclusions”. It is possible to suspect the aim, but it should be formulated more precisely. The main conclusion is not highlighted at the end of Introduction according to the demand.

Our response: Thanks for your constructive advice. According to the demand of the Journal, we have made corresponding changes, please see it in the revised version (lines 87-91).

Comment 4: I do not understand the use of several literature sources. For instance, authors write in lines 71 and 72 “A recent study found that the application of P, K, Zn fertilizer combined with 70 low-level application of N and Fe could significantly reduce bean powdery mildew disease [26].” When I look for the source [26] in the list of literature, I find that the topic of this article is on disease of tomatoes, and not on bean powdery mildew ( Weerasinghe, K.M.S.; Balasooriya, A.H.K. Effect of Macro and Micro nutrients on incidences of the Blossom End Rot in Tomato ) .

Similarly, very good papers [4–5] are used to tell that “Cotton, a primary natural fiber crop, is of great importance to the global textile industry [4]. Unfortunately, China suffer from yield losses of cotton were tremendous in V. dahliae-infested soil annually [5]” (lines 40–42). But the topics of those papers are as follows “Physiological and molecular mechanism of defence in cotton against Verticillium dahliae [4], and “An overview of the molecular genetics of plant resistance to the Verticillium wilt pathogen Verticillium dahliae. [5]. This makes me think that some articles are not used according to their topic, and references in the Introduction must be reconsidered one more time.

Our response: Yes, it is our fault, we have carefully checked the all references and changed 4, 5, 26, 27, 46 references.

Comment 5: As section Results is put before the Materials and Methods section, abbreviations used should be decoded in this section, e.g. “dpi”, “hpt”, “DI”.

Text in lines 167–174 should be justified.

Our response: Yes,“dpi”, “hpt”, “DI” abbreviations have been decoded when they first appeare, and text in lines 167–174 has been justified.

Comment 6: Do not forget the Latin names of living organisms, which are mentioned for the first time in this section. As the aim is not formulated precisely and understandably, I cannot judge the conclusions. To me they seem unclear and too wordy at the final sentences of the discussion. In my opinion, conclusions are better formulated in the Abstract.

Our response: We really appreciate your advice, the main aim and conclusion of this study were added in “Introduction” section. In addition, we try our best to accurately describe the conclusion, please see lines 232-234, 246-247, 258-259.

Comment 7: The demand of the Journal is as follows: “SI Units (International System of Units) should be used”,but rpm is not a SI unit. Even if the Journal agrees that you use this unit, which is not easily and understandably transformable to SI system unit, you need to take into account equal writing style; now you have rpm (e.g., line 294, 317) and r/min (e.g. line 297).

You can use kg/hm2 , but I suggest to use kg/ha; SI System allows to use ha.

Our response: Yes, SI Units should be used, r/min and kg/ha were unified used in the full text.

Comment 8: It would be nice to see more details on equipment used, e.g. hemocytometer (line 298) microscope (line 312) – please, let us know the trade mark and producer; for microscope – what magnification is used.

Our response: Yes, the trade mark and producer of hemocytometer and microscope were added, and the microscope used in this study is a 40 X eyepiece (line 315).

Comment 9: In line 331–332, you write about calculation of disease index, and reader can only see the reference [53] here. I suggest to make a brief description, to show formula, and then put reference.

Our response: As you suggest, we have added detailed information about disease index, please see lines 335-339.

Comment 10: Every source is doubly numbered. Description of some sources is not completed, e.g. sources [26] (lines 447–448), source [46] (line 496); check, please, also others for correctness.

Our response: We have carefully checked the all references and revised every source to be single numbered. Thanks again.

Comment 11: Some spelling mistakes are found, e.g., line 138: ‘plamts’; line 220: ’pathgenic’, line 249 ‘Xylem’ started with a capital letter in the middle of sentence.

Our response: Yes, we have corrected the spelling mistakes. Please see lines 140, 222, 251.

Comment 12: I also suggest to not start a sentence with a numeric number, e.g. instead of “100 μL of the microsclerotia...” write “One hundred millilitres of the...” (Line 309). Look at similar cases also.

Our response: Yes, thanks for your suggestion, we have revised the description (lines 303, 312, 319).